# Non-contact identification and differentiation of illicit drugs using fluorescent films

Ke Liu[1], Congdi Shang[1], Zhaolong Wang[1], Yanyu Qi[1], Rong Miao[1], Kaiqiang Liu [1], Taihong Liu[1] & Yu Fang[1]

Sensitive and rapid identification of illicit drugs in a non-contact mode remains a challenge for years. Here we report three film-based fluorescent sensors showing unprecedented sensitivity, selectivity, and response speed to the existence of six widely abused illicit drugs, including methamphetamine (MAPA), ecstasy, magu, caffeine, phenobarbital (PB), and ketamine in vapor phase. Importantly, for these drugs, the sensing can be successfully performed after $5.0 \times 10^5$, $4.0 \times 10^5$, $2.0 \times 10^5$, $1.0 \times 10^5$, $4.0 \times 10^4$, and $2.0 \times 10^2$ times dilution of their saturated vapor with air at room temperature, respectively. Also, presence of odorous substances (toiletries, fruits, dirty clothes, etc.), water, and amido-bond-containing organic compounds (typical organic amines, legal drugs, and different amino acids) shows little effect upon the sensing. More importantly, discrimination and identification of them can be realized by using the sensors in an array way. Based upon the discoveries, a conceptual, two-sensor based detector is developed, and non-contact detection of the drugs is realized.

---

[1] Key Laboratory of Applied Surface and Colloid Chemistry of Ministry of Education, School of Chemistry and Chemical Engineering, Shaanxi Normal University, Xi'an 710062, People's Republic of China. Correspondence and requests for materials should be addressed to Y.F. (email: yfang@snnu.edu.cn)

llicit drugs, especially the synthetic ones, have posed serious threat to human health, family harmony, and social stability[1,2]. Thus, development of sensitive, selective, rapid, and inexpensive methods for in situ detection of illicit drugs is of great significance for preventing and reducing drug-related crimes. Although various techniques, such as gc-mass method[3,4], ion-mobility spectrometry[5], surface-enhanced Raman spectrometry[6,7], electrochemical luminescence[8], and fluorescence techniques[9,10] have been developed to detect drugs, no methods can be compared to specially trained animals. Even worse, the reported techniques or instruments are hard to be used in situ since pretreatment and/or pre-concentration of the samples need to be conducted before any valuable detection, which explains why on-site screening of illicit drugs still relies on drug-sniffing dogs and drug-suppressing police officers.

Modern optical or electrical technology-based vapor detection techniques have been expected to work instead of sniffer dogs[11–15]. As an example, hidden explosive detection via vapor phase sensing has been realized by using ion-mobility and fluorescence techniques[16–18]. Compared with other known methods, film-based fluorescence sensing possesses several unique advantages, such as great designability, outstanding sensitivity, reusability, and low cost. Because of such reasons, research in the field has achieved great progress during the last few decades[19–25]. Studies in the detection of illicit drugs with this technology, especially in vapor phase, however, are limited. Cheng and co-workers[10,26–28] reported several fluorescence methods for the detection of methamphetamine (MAPA) and ketamine, and the lowest detection limit (DL) in vapor phase of MAPA is ~180 ppb, and 50 pg cm$^{-2}$ for ketamine in aqueous phase. Recently, Pavel and co-workers[29,30] reported a supramolecular sensor array composed of fluorescent cucurbit[n]uril-type receptors, which enables the detection of pseudo-ephedrine and opiates as well as their metabolites in human urine. The DLs of the detections are lower than those reported for solid phase extraction-high-performance liquid chromatography. Based on the same principle, Kim and co-workers[31] reported a strategy that combines the selectivity of supramolecular chemistry and the sensitivity of organic field-effect transistors to detect MAPA at a low concentration (~1 nM) in human urine. Reviriego et al.[32] discovered that the sodium salt of diethyl-1H-pyrazole-3,5-dicarboxylate could function as a receptor of MAPA. Based upon this finding, they designed a sensitive fluorescence sensor for MAPA, amphetamine, and ecstasy in solution phase. Recently, we[9] reported a fluorescent sensor for MAPA and its simulant, N-methyl-phenethylamine, which shows a DL of ~5.5 ppb in vapor phase. However, all the reports either have their own limitations, as types of drugs detected are limited, or the studies remain at a principle level.

The reasons behind the slow progress in the research, especially detection in vapor phase, might be the following: (1) most of the drugs concerned exist in hydrochloride or sulfate form, which will greatly limit their vapor pressure; and (2) from the viewpoint of practical uses, incompatibility between universality for sensing different illicit drugs and selectivity to avoid interferences is an even bigger challenge for developing the detection methods and related detectors. A reasonable route to solve these problems is to develop individual high-performance illicit drug sensors first, and then combine them into a sensor array[33–37]. To be more compact, simply structured and have less power consumption, the component sensors in an array should work in a similar way.

Perylene bisimide derivatives (PBIs) are a group of exceptional fluorophores for detecting amido-bond-containing organics, not only because of their extraordinary photochemical stability but also because of their appropriate energetics that guarantees electron transfer from the analytes to the photoexcited PBIs[38–40]. As most illicit drugs contain amido bond, PBIs may be adopted as

suitable sensing fluorophores in the development of the films. However, PBIs tend to aggregate and form fluorescence silent H-aggregate, which is unfavorable for sensing. Therefore, PBI needs to be modified to hinder the unfavorable aggregation[41–43]. In addition, substrate could also play roles as it affects the affinity of the analyte molecules to the sensing units and alternates the structures of the films[44–46]. Accordingly, new films can be created by simply changing the substrates while the sensing fluorophores and the fabrication strategies are kept unchanged. No doubt, the differences originated from the change will contribute to the identification and discrimination of the analytes.

Here an o-carborane derivative of PBI (PBI-CB), of which o-carborane occupies a nonplanar structure, is designed and synthesized. As expected, the fluorescence quantum yield of the compound in solid state is significantly higher than that of a routine PBI dimer (PBI-PE). Moreover, PBI-CB shows better solubility in common organic solvents and exhibits rich self-assembly property, forming nanofibers in suitable medium. Combination of the PBI-CB fibers with different substrates, three fluorescent films have been fabricated. Interestingly, the films as created show different responses to the vapors of six commonly found illicit drugs over a variety of potential interferences. Then a two-step judgment strategy (Fig. 1a) is proposed and the sensors are combined into a sensor array to satisfy the diverse practical requirements and to identify the drugs under examination. Particularly, the drugs in their hydrochloride forms with extremely low vapor pressure can be directly detected, a result never reported before.

## Results

**Fluorescent behavior of PBI-CB**. Compared with a route PBI dimer (PBI-PE), the newly synthesized nonplanar PBI dimer, PBI-CB (Fig. 1b), emits much brighter red fluorescence under ultraviolet (UV) light in solid state (Fig. 1c), suggesting that the PBI units in the solid should exist in J-type aggregates[47]. In contrast, the dark red color and less-emissive property of solid-state PBI-PE is an indication of H-type aggregation of its PBI units. The differences in fluorescence behaviors of the two PBI dimers clearly demonstrate that as expected, introduction of o-carborane unit is an effective way to screen H-type aggregation of PBI units owing to steric hindrance.

To further understand the differences between the two PBI dimers, their absorption and emission spectra in dioxane, toluene, diethyl ether, and cyclohexane were recorded, and the results are depicted in Fig. 1d. With reference to the figure, it is seen that the UV-vis spectra of PBI-CB in dioxane and toluene are characterized by three well-resolved vibronic bands, and at the same time, the intensities of the bands increase from 0–2 to 0–1, and then to 0–0, which is a typical monomeric absorption of PBI, suggesting PBI-CB dissolved monomerically in the solvents[48]. This is consolidated by the fine structures of corresponding fluorescence emissions as depicted in the same figure. However, when diethyl ether and cyclohexane were used as solvents, the profiles of the absorption and emission spectra of the compound changed greatly. As shown in the figure, the three absorption bands became less resolved, and their relative intensities changed as well, which are strong evidence of aggregation of the PBI unit of the compound in the solvents. Broad and redshifted emission further confirmed this statement[49].

Interestingly, for the control compound, PBI-PE, similar absorption and emission behavior in dioxane and toluene was observed (Supplementary Fig. 1), suggesting its monomeric nature in the solvents. For the solvents of diethyl ether and cyclohexane, however, no obvious emission was found even though it shows similar absorptions in the systems under study, a

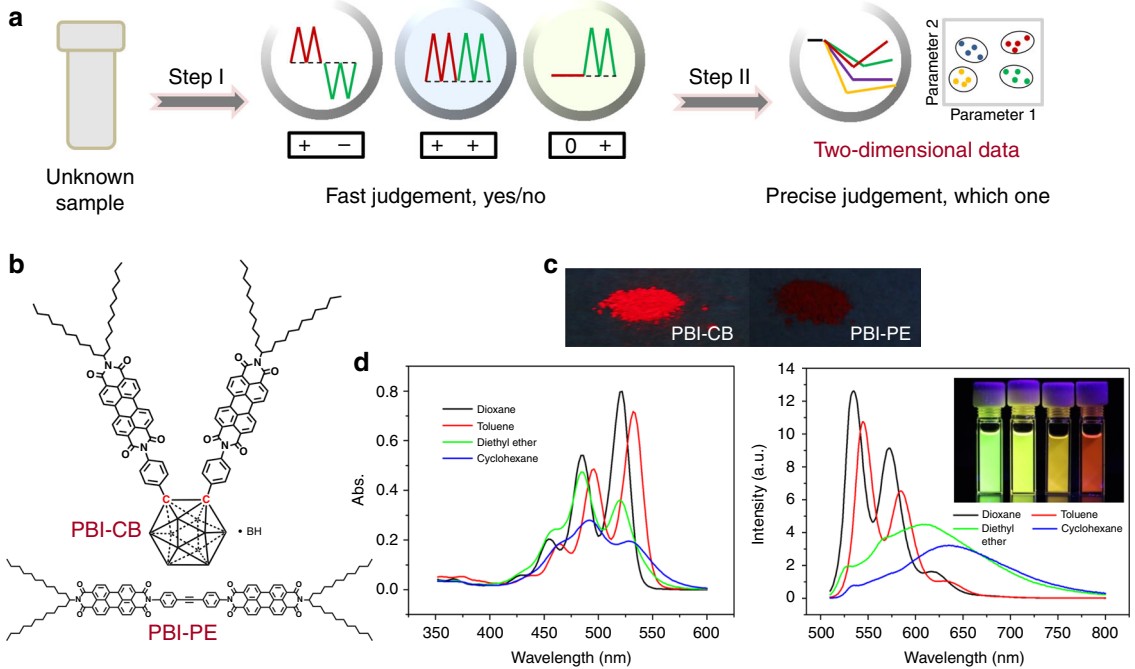

**Fig. 1** Analytical strategy and fluorescent property of PBI-CB. **a** Schematic representation of the two-step judgment strategy for differentiation and identification of illicit drugs via scent sniffing. **b** Chemical structures of PBI-CB and PBI-PE. **c** Fluorescent images of PBI-CB and PBI-PE in solid state under UV light (365 nm). **d** UV-vis absorption spectra and fluorescence emission spectra of PBI-CB recorded in different solvents at a concentration of $5.0 \times 10^{-6}$ mol L$^{-1}$ and at room temperature ($\lambda_{ex} = 480$ nm). Inset: fluorescent pictures of the four solutions under UV light (365 nm)

strong indication of H-type aggregation of the PBI unit of the compound[42]. These results once again demonstrate that the aggregation behavior of PBI unit can be effectively tuned by employing stereo-effect at a molecular level, laying foundation for designing high-performance fluorescent sensing systems.

**Aggregation behavior of PBI-CB in solvent**. As revealed in fluorescence behavior study, the designed PBI dimer, PBI-CB, is always highly emissive no matter dissolved in good solvents or poor solvents or existing in solid state. This unusual behavior is ascribed as a result of the unique stereo-structure of the compound, which blocks H-type packing of the PBI unit during aggregation of the compound. Accordingly, highly emissive fluorescent films could be fabricated using solution casting method. Based upon this precious property, three fluorescent PBI-CB films were fabricated, of which plastic plate, bare glass slide, and silica-gel plate were used as substrates, respectively, and the corresponding films are respectively named as film 1, film 2, and film 3.

To have a better understanding of the aggregation behavior of the PBI derivative under study, transmission electron microscopy (TEM) measurements were performed to examine the morphologies of the self-assemblies of PBI-CB obtained at different times from a mixture solvent of dioxane and water (92.5%/7.5%, v/v) at a concentration of $5 \times 10^{-5}$ mol L$^{-1}$. The results are shown in Fig. 2. Clearly, the morphologies of the aggregates are time-dependent. At the beginning ($t < 1$ h), the compound aggregated into nanoparticles of an average diameter of ~6 nm, then the particles merged into primary fibers ($1$ h $< t < 3$ h) with extremely high length–width ratio, and finally these fibers aggregated into larger fibers as revealed by TEM and scanning electron microscopy studies (Supplementary Fig. 2a, c). Interestingly, the widths of the primary fibers are almost the same, and equal the average diameter of the nanoparticles. Moreover, the fibers kept constant during the whole self-assembly process even though they

tend to further aggregate, suggesting the nanoparticles formed are anisotropic as they grow and aggregate in well-defined directions. This statement is confirmed by the layered structures of them as revealed by electron diffraction and X-ray diffraction studies (Supplementary Fig. 2b), of which the thickness of the layer is ~6 nm.

It is anticipated that it is the fibrous structures that would make the building block, the sensing fluorophore PBI-CB, expose to air, which would increase the opportunity for the analyte molecules to reach them, resulting in faster response when they are used for sensing. Moreover, selectivity may also be improved owing to hydrophobicity of the fibers.

**Preliminary identification of illicit drugs**. Before sensing performance studies, the steady-state fluorescence excitation and emission spectra of the films were recorded, and the results are depicted in Fig. 3a. It is seen that the profiles of the excitations and emissions of them are slightly different from each other, but the maximum excitation and emission wavelengths are the same, which are 480 and 650 nm, respectively. As expected, the emissions are characterized by a broad band, a typical PBI aggregate emission. In addition, large Stokes' shift (~170 nm) and strong emission as demonstrated would be favorable to further sensing studies.

As it is well known, for real-life application, simple operation, interference-free, and analysis with real-world samples are key criterions to evaluate a sensing method or sensing device. Accordingly, in addition to widely consumed illicit drugs, including MAPA, magu, ecstasy, ketamine, caffeine, and phenobarbital (PB), water, shampoo, cream, hair conditioner, apple, pear, banana, dirty clothes, and a variety of amido-bond-containing organic compounds were taken as interferences to conduct the tests. Water is chosen for study as it frequently shows quenching effect to the emission of fluorescent films[50–52]. Similarly, amido-bond-containing compounds are taken as they

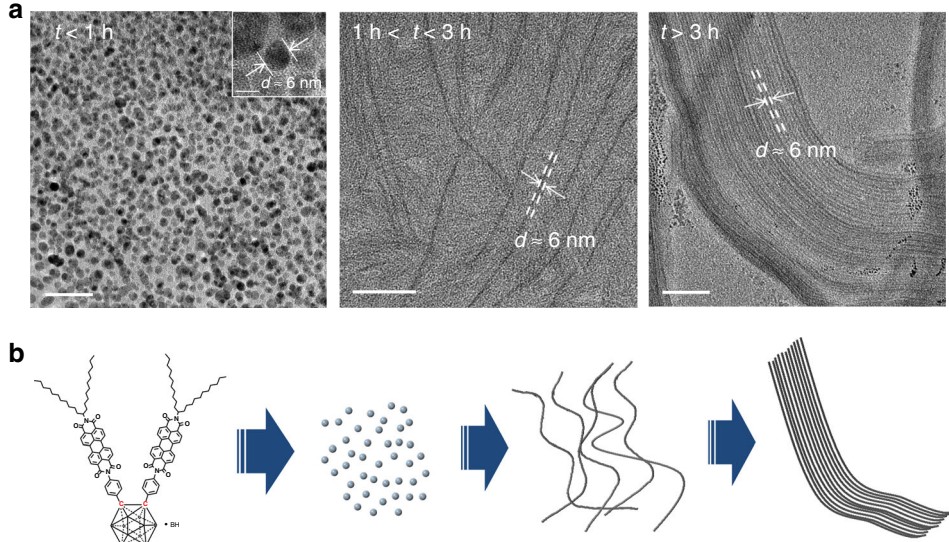

**Fig. 2** Self-assembly of PBI-CB in solution. **a** TEM images of the self-assembled structures of PBI-CB at different times in a mixture solvent of dioxane:water ($V:V = 92.5:7.5$). It is seen that initially, the molecule of PBI-CB self-assembled into nanoparticles ($t < 1$ h), then nanofibers ($1$ h $< t < 3$ h), and finally nanoribbons ($t > 3$ h), scale bar: left, 20 nm; the insert of the left, 5 nm; middle and right, 100 nm. **b** Schematic representation of self-assembly process of PBI-CB

are the most commonly encountered quenchers of PBIs[38–40]. Considering practical illicit drug detection, it is reasonable to take real usage scenarios into account. As well known, for sniffer dogs, false alert is frequently induced by odorous substances in travelers' luggage, such as toiletries, fruits, and even dirty clothes, which is the reason why they have also been chosen as interferences even though they are rarely mentioned in earlier studies[53].

It is to be noted that the samples and the potential interferences were used directly without any pretreatment, including pre-concentration. Besides, with the exception of spectroscopy measurements, all sensing tests were conducted on a homemade single-sensor-based sensing platform (Fig. 4). Different from commercially available instruments, this platform was specially designed to work in a way to satisfy the requirement of practical uses. For example, (1) sampling in two different ways, which are wiping and sucking, (2) reporting results on line, (3) light-emitting diode as light source, (4) photodiode as the emission detector, etc. Therefore, with suitable film device, the apparatus can be taken as a conceptual illicit drug detector. To perform the test, the fluorescent films to be examined were made into devices first. Figure 4a shows a schematic structure of the sensor, and the inset of Fig. 4b is a picture of a representative film device. The results from the tests are depicted in Fig. 3b–e and Supplementary Fig. 4.

Overall, the three films exhibited different response behaviors to the illicit drugs and the interferences examined. Specifically, for film 1 and film 2, the illicit drugs under tests showed quenching effect to the film's emission with different quenching efficiencies (Fig. 3b, c). For film 3, however, they showed no significant effect (Fig. 3d). As for the examined interferences, their effect upon the film's emission can be grouped into four categories, of which one is water and six odorous substances, which are the non-pure chemicals with dirty clothes as one only exception; the second is dirty clothes, glycine, and dopamine; and the third is the others excluding benzylamine, ethylhexylamine, and piperidine, which all belong to group four (Fig. 3, Table 1).

With reference to the figures and the table, it is seen that the first group of the interferences demonstrated quenching effect upon the fluorescence emission of film 1 and film 3, but sensitization to that of film 2. Group 2, however, showed

negligible effect upon the emission of the three films. Group 3 showed no observable effect upon the emission of film 1 and film 2, but small quenching effect upon film 3. As for group 4, they are quenchers of the emission of film 1 and film 2. But the effect to the emission of film 3 is complicated. Specifically, two of the three are sensitizers, and one is a quencher.

The results as described reveal great effect of the substrates upon the sensing behaviors of the films even though they possess exactly the same sensing fluorophore. Importantly, the differences in the sensing behaviors of the three films provide a chance to discriminate and identify the illicit drugs under study.

Considering the judgment criteria aforementioned, two of the three films, which are film 1 and film 3 or film 2 and film 3, under examination can be grouped into a logic gate to identify an unknown sample. First, truth table was generalized from the consequences of the sensing tests (Table 1a). Both sensitization and quenching are counted as "1", no response as "0". Based upon the truth table, two logic gates can be constructed as depicted in Table 1b. In this way, preliminary but reliable identification of the illicit drugs can be realized within 10 s. This is because the response of the films is fast, and the repeatability of the tests is excellent.

**Discrimination of illicit drugs**. Discriminative and selective detection of the illicit drugs under examination were realized by conducting real-time and in situ analysis of the information acquired from the film-based sensing tests. As discussed in the preceding paragraph, analysis of the information embedded in Fig. 3 and Table 1 could tell if there is an illicit drug. For practical uses, however, it is better to make sure which drug it is. Thus, to identify the drug, the response dynamics of film 1 to the target drugs were examined.

To realize the discrimination, every drug sample was tested 15 times, and the results are depicted in Supplementary Fig. 4. It is seen that there are two stages in the responses of the films upon each sampling of the drugs as reflected in the response traces, which are records of intensities against time. It is seen that upon each sampling of the drugs, the fluorescence emission of the film first decreases, and then the emission stops to decrease or even

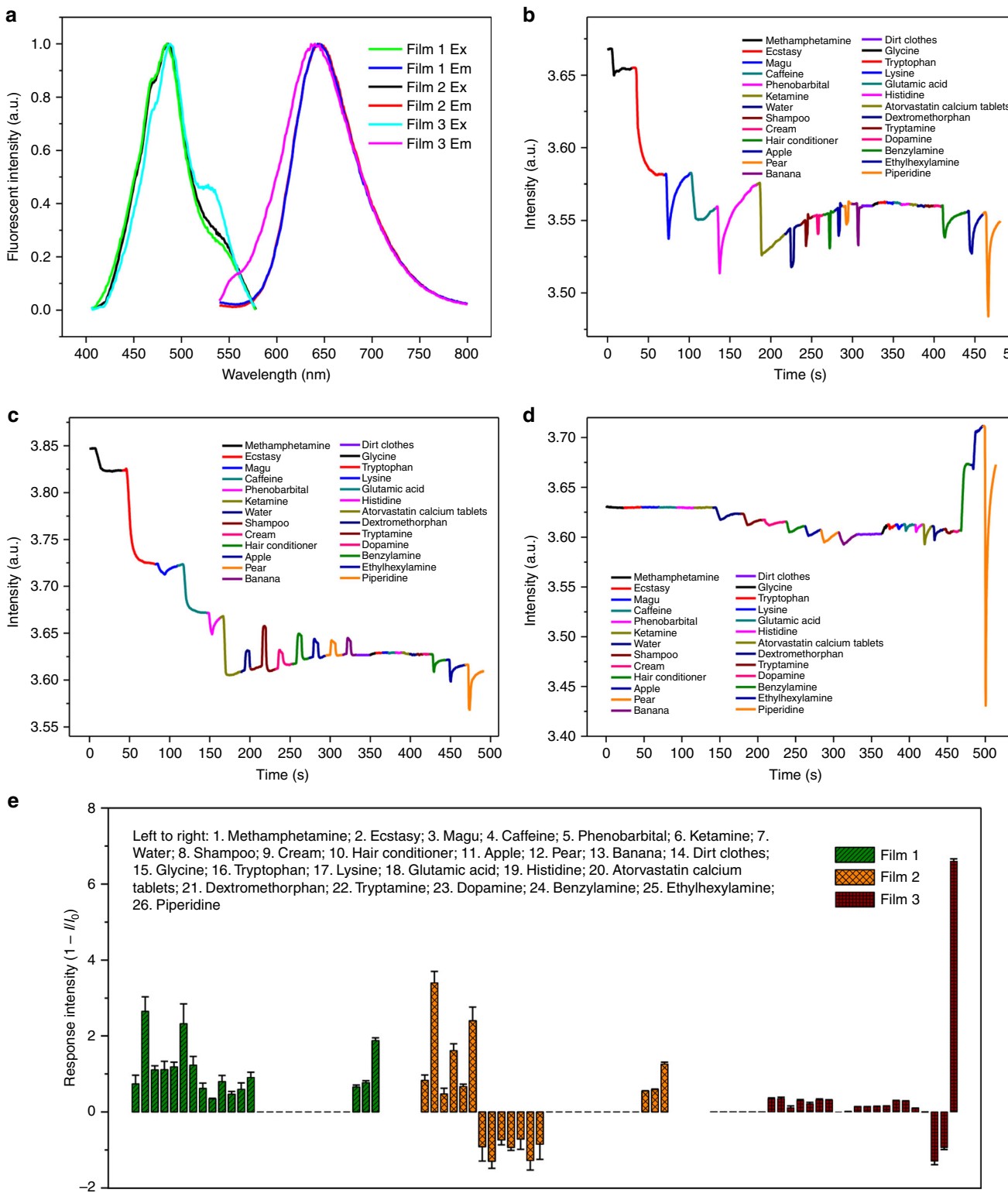

**Fig. 3** Fluorescence spectra and sensing properties. **a** Fluorescent excitation and emission spectrum of the films ($\lambda_{ex}/\lambda_{em}$ = 480/650 nm). Fluorescent intensity of film 1 (**b**), film 2 (**c**), or film 3 (**d**) monitored at 650 nm ($\lambda_{ex}$ = 480 nm) vs. time with exposure to different illicit drugs and different potential interferences. **e** Collective display of the sensing properties of the three films from five repetitive measurements. Note: (1) it is seen that different films exhibit different response behavior to the analytes, and at the same time different analytes show different effects upon the fluorescence emissions of the films; (2) the results depicted in **b**–**d** are typical one of the five repetitive tests; (3) the illicit drug samples tested are MAPA, ecstasy, magu, caffeine, phenobarbital and ketamine, and the potential interferences examined include water, toiletries (shampoo, cream, and hair conditioner), fruits (apple, pear, and banana), dirty clothes, and some amido-bond-containing organic compounds; (4) the error bars are drawn from the maximum responses

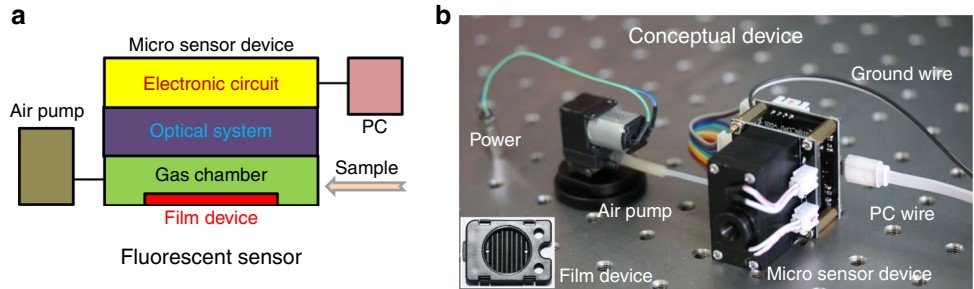

**Fig. 4** Fluorescent sensors and sensing platform. **a** Schematic representation of a fluorescent sensor. **b** One-sensor-based sensing platform for evaluation of sensing properties of fluorescent films

---

**Table 1 Discrimination and identification of the drugs**

a

| Film 1 | Film 2 | Film 3 | Analytes |
|--------|--------|--------|----------|
| 0 | 0 | 0 | No |
| 1 | 1 | 0 | Illicit Drugs |
| 1 | 1 | 1 | Interferences-1 |
| 0 | 0 | 0 | Interferences-2 |
| 0 | 0 | 1 | Interferences-3 |
| 1 | 1 | 1 | Interferences-4 |

b

Film 1 / Film 3

$F = A \cdot \overline{B}$

Film 2 / Film 3

$F = A \cdot \overline{B}$

| A | B | F |
|---|---|---|
| 0 | 0 | 0 |
| 1 | 0 | 1 |
| 1 | 1 | 0 |
| 0 | 0 | 0 |
| 0 | 1 | 0 |
| 1 | 1 | 0 |

a The truth table of the responses recorded as shown in Fig. 3, where positive response no matter quenching or sensitizing is counted as "1", and no response as "0". b The logic gates formed with any two of the three films
Note: interferences-1: 7–13; interferences-2: 14, 15, 23; interferences-3: 16–22; interferences-4: 24–26
1. Methamphetamine; 2. ecstasy; 3. magu; 4. caffeine; 5. phenobarbital; 6. ketamine; 7. water; 8. shampoo; 9. cream; 10. hair conditioner; 11. apple; 12. pear; 13. banana; 14. dirt clothes; 15. glycine; 16. tryptophan; 17. lysine; 18. glutamic acid; 19. histidine; 20. atorvastatin calcium tablets; 21. dextromethorphan; 22. tryptamine; 23. dopamine; 24. benzylamine; 25. ethylhexylamine; 26. piperidine

---

starts to recover when the sampling is stopped. However, the traces as recorded for different drugs are different from each other. Specifically, for magu, the film's emission decreases within the whole test even though the quenching speed is getting slower upon removing the sample, and for the sensing of MAPA, caffeine, and ecstasy, the fluorescence emission stops to decrease when the samples are removed. It is to be noted that at these two cases and within the <50 s timescale no significant fluorescence recovery was observed. For ketamine and PB, full recovery of the initial emission can be achieved. Therefore, from the different response dynamics of film 1, the six drugs may be discriminated provided sufficient data were collected.

With the results shown in Fig. 5, an accurate discrimination of the six drugs can be simply made. To conduct the discrimination and identification, a two-step strategy is utilized, of which the first step takes <10 s (Fig. 5a). Within this step, combined use of the information obtained from two of the three films will provide a "yes" or "no" answer, telling if the sample is a target illicit drug or not. For the systems of a "yes" answer, startup of the second step is needed, which takes a longer time but not more than 1 min.

The key content of the second step is a detailed analysis of the response kinetics of the film to the drugs under examination. Figure 5b schematically shows the responses of film 1 to the presence of the illicit drug vapors. To be easier for operation, a parameter, R, is introduced, which is defined as

$$R = (I_b - I_c)/(I_a - I_b) \tag{1}$$

where $I_a$, $I_b$, and $I_c$ represent the relative fluorescence intensities of the film at the start of sampling, at the end of the sampling, and at a certain time after the removal of the sample, respectively. It is to be noted that the sampling process usually takes <10 s, and the recovery <50 s. Comparison of the R value reveals that the film-to-drug response can be grouped into three types, of which one is $-0.2 < R < 0.1$, that is the fluorescence intensity of the film stops to decrease after the drug is removed. MAPA, caffeine, and ecstasy conform to this type. For type two ($R > 0.3$), the fluorescence emission continues to attenuate after removing the drug. Of the six drugs, only magu belongs to this type. For the third type ($R < -0.8$), the fluorescence emission starts to recover after the drug is removed. Ketamine and PB conform to this type. With further examination of the attenuation and recovery traces, it is also seen that the response kinetics are different from one another within type one or type three, laying a foundation for further discrimination. To conduct the discrimination, different methods were employed. For type one, both the trace for quenching stage (a to b) and that for recovery stage (b to c) were fitted with Eq. 2. In this way, two parameters $A_1$ and $A_2$ for each test could be obtained, which stands for a point in Fig. 5c. Clearly, 15 tests for each drug in the type fall in the same cluster, and the three clusters are well separated, indicating a clear discrimination between them. For type three, Eq. 3 was used to fit the traces of the quenching process (a to b), and two parameters A and B were generated. Again, the two drugs belonging to this type can also be well separated. The results depicted in the figure demonstrate a strong discrimination and identification capability of the virtual

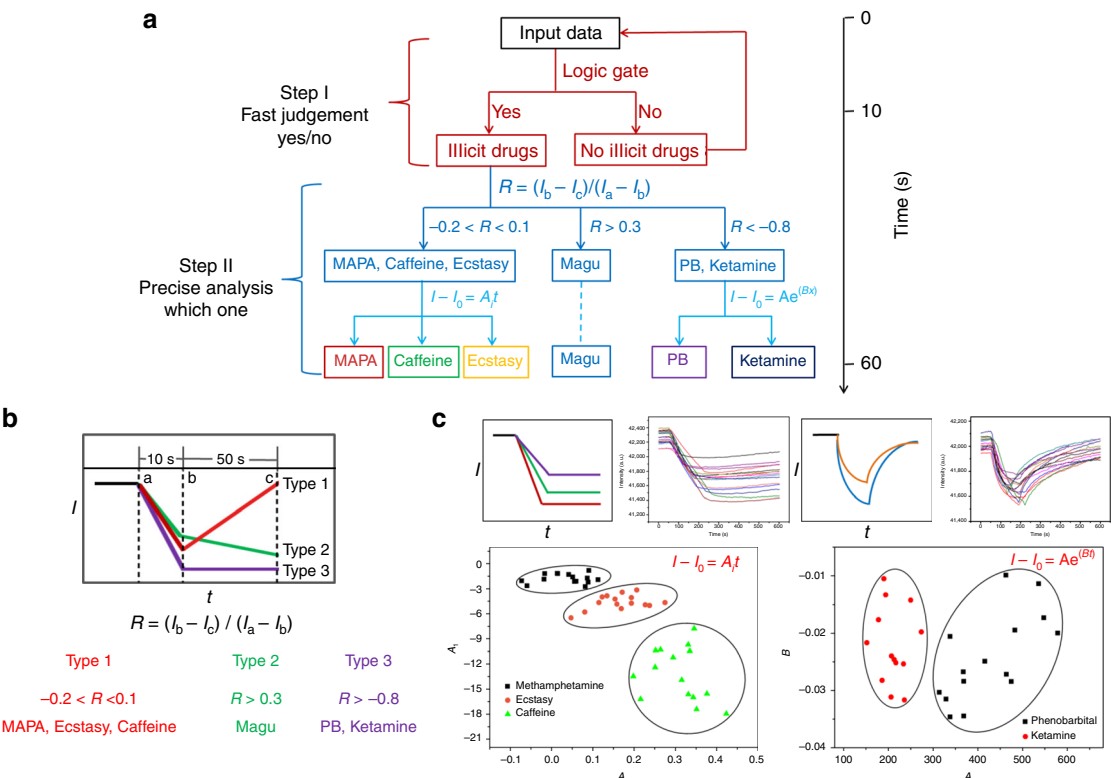

**Fig. 5** Two-step discrimination of illicit drugs. **a** Logical judgment process to the presence of unknown sample via utilization of three fluorescent films, where a two-step strategy is used to identify and differentiate the samples under examination. In the first step, which takes <10 s, combined use of two of the three films will provide a "yes" or "no" answer, which tells if the sample is a target drug or not. In the second step, which takes a longer time but not more than 1 min, there are two stages to differentiate the chemical nature of the found drug in this step. **b** Schematic representation of the dynamic responses of film 1 to the presence of different illicit drugs in vapor phase, of which the responses can be qualitatively grouped into three types. **c** Discrimination of the illicit drugs under examination, where a parameter $R$ was introduced, which can be calculated in the way as depicted in **b**. Clearly, the detectable six drugs can be simply and efficiently identified, indicating the strong discriminating ability of the film. It is to be noted that the parameters $R$, $A_1$, $A_2$, $A$, and $B$ for the illicit drugs were calculated from the traces of the 15 repetitive tests (Supplementary Fig. 4), and the detailed information is depicted in Supplementary Tables 2–4

sensor array to the drugs examined.

$$I - I_0 = A_i t \qquad (2)$$

$$I - I_0 = Ae^{(Bt)} \qquad (3)$$

where $I$ and $I_0$ stand for the relative fluorescence intensities of the film at time $t$ and time a ($t_a$) or time b ($t_b$), respectively.

**Two-sensor-based sensing apparatus**. With careful examination of Figs. 3 and 5 and Table 1, it can be seen that the information obtained from film 2 is just used for confirming the judgment of "yes" or "no" made in the first step of the two-step strategy, and analysis in the second step utilizes only the information obtained from film 1, suggesting that fast analysis could be realized with collective using of film 1 and film 3 only. Accordingly, a conceptual detector was built with the two-film-based device as the main component (Supplementary Fig. 5a). With this detector, sensing tests can be conducted by using the two films simultaneously, resulting in faster sensing, lower energy consumption, and more compact instrument structure.

Supplementary Fig. 5b depicts the results from determination of three typical illicit drugs (MAPA, magu, and caffeine), water, and two odorous substances, shampoo and apple. Clearly, the response traces from the two-film sensors are nearly the same with those obtained independently by the sequential

use of them (Fig. 3b, d). Based upon the results, a logic gate (Supplementary Table 1) was generated, which tells, as expected, whether the sample under examination is an illicit drug or not. As for the second step in the strategy, same operation can be performed with this two-sensor-based sensing apparatus.

**Theoretical calculations**. The quantum chemical methods at the B3LYP/6–31G (d)[54] level of theory using Gaussian 09 were employed for the calculation of the highest occupied molecular orbital (HOMO) and lowest unoccupied molecular orbital (LUMO) energies for optimized geometries of PBI-CB and the relevant drugs, including PB, caffeine, ketamine, ecstasy, and MAPA, of which magu is not included as its main content is MAPA. The hole and particle energies of PBI-CB, and the HOMO energy of an example drug, MAPA, are presented in Fig. 6a. The HOMO and LUMO energies of the fluorescent compound and all the relevant illicit drugs are depicted in Fig. 6b.

Reference to the information shown in Fig. 6b reveals that all the five drugs as examined are all suitable quenchers, at least energetically, as their energies of the HOMO orbitals are higher than that of PBI-CB. These differences ensure the hole of the fluorophore after excitation can accept an electron from the drug molecule, the so-called photoinduced electron transfer (PET), resulting in the observed fluorescence quenching[55,56]. Figure 6a schematically shows the electron transfer from

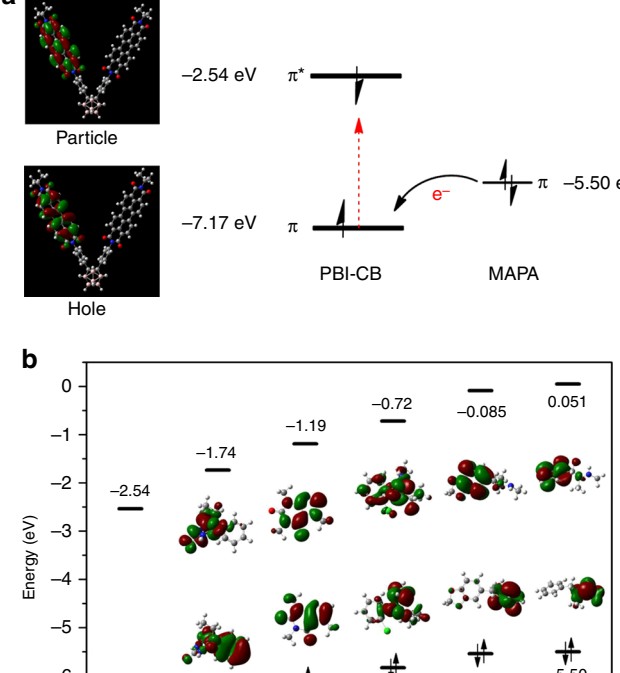

**Fig. 6** DFT calculations of PBI-CB and the drugs. **a** The electron density distributions of the fluorescence emission relevant orbitals of PBI-CB, and possible quenching mechanism in the presence of the tested illicit drugs. **b** Dominant natural transition orbital pairs for the lowest singlet excited state geometry, and levels of HOMO ($\pi$) and LUMO ($\pi^*$) orbitals of PBI-CB as well as the relevant illicit drugs

the HOMO orbital of MAPA to the hole of PBI-CB after excitation.

## Discussion

As aforementioned, unlike commonly found PBI chemical dimers, the synthesized o-carborane-mediated PBI dimer, PBI-CB, shows strong fluorescence emission in solid state. The profile of the emission lacks fine structures and appears at longer wavelengths than that of the monomeric PBI emission, suggesting that PBI unit of the compound exists in aggregated but not fully overlapped $\pi$–$\pi$ packed state that is J-aggregated state. This unusual behavior can be ascribed to the unique structure of o-carborane. This is because the tethering of PBI to the C1–C2 position of o-carborane ruled out its possibility to exist in a planar conformation owing to the three-dimensional structure of o-carborane, which represses fully overlapped $\pi$–$\pi$ stacking between different PBI units both intermolecularly and intramolecularly[57,58]. As a control, PBI-PE, however, emits very weak in solid state, which is a result in further support of the conjecture of hindering of H-aggregate formation of PBI-CB in solid state. The steric hindrance introduced by o-carborane is also beneficial for sensing as the films formed by the compound must contain rich of micropores or channels, which are essential for fast and reversible sensing owing to diffusion of analyte molecules within the films[59,60].

As demonstrated already, the sensing behaviors of the three films are different from each other. The reasons behind could be

either the fluorescence behavior determined by the structure of PBI-CB aggregate on the film substrate or the property of the substrates themselves. Comparison of the steady-state fluorescence emission spectra of the films reveals that there is no significant difference between them, indicating that PBI-CB aggregated in a similar way on the three substrates employed. The surface structure and property of the three substrates, however, are different from each other. For example, the surface of silica-gel plate is rough, but the surfaces of the other two are smooth, and the surface of plastic substrate is hydrophobic, whereas the surfaces of the others are hydrophilic, which may explain the differences between the sensing behaviors of the films.

The high sensitivity, fast response, and multiple illicit drug detection of the films as fabricated should be ascribed to the uniquely designed structure of PBI-CB. First, it matches, energetically, the requirement of the drugs for PET-based sensing. Second, the geometrical structure of PBI-CB is conducive to its sensing applications in both solid and aggregated state, as it screens H-type stacking of the PBI units and, at the same time, favors formation of fibrous structures at a molecular level via molecular packing.

To our surprise, film 1 and film 2 as created could detect the six illicit drugs in their original states via vapor phase sampling (10 s). This is astonishing because as real-world samples of the drugs, MAPA, magu, ecstasy, and ketamine exist in salt forms, which means their vapor pressures are much lower than those in neutral states (Fig. 7a). For MAPA, magu, and ecstasy, treatment of the samples with a base, such as $Na_2CO_3$, resulted in more than 10 times enhancement of the responses (Fig. 7b). The enhancement was also observed for ketamine, but the improvement is limited. The reason behind might be the hydrophobic environment of the ionic center, which could potentially screen the approach of the inorganic base.

The interferences, in particular the odorous substances aforementioned, are prone to induce false positives in sensing illicit drugs when using drug-sniffing dogs and other common techniques, including fluorescence. Therefore, ruling out their interference is the prerequisite to a successful detection. To our luck, the films as developed showed different response behaviors to the illicit drugs and the broad and potential interferences, laying a foundation for developing them into practical detectors. The different behaviors of the three films upon exposure to the vapors of the illicit drugs or the interferences could be ascribed to the different surface structures and properties, such as roughness, hydrophobicity, etc., which may screen the interferences or illicit drugs owing to compatibility reasons. Surely, energetics and volatility of the samples and interferences may also play important roles.

In real-life application, sensitivity is another important factor. Therefore, the DLs of the films to the $Na_2CO_3$-treated drugs were determined with the homemade conceptual sensing device as a study platform. The DLs of the illicit drugs, which were represented by the maximum dilution ratios of air to the equilibrium vapor of the drugs (volume to volume), for MAPA, ecstasy, magu, caffeine, PB, and ketamine are $5.0 \times 10^5$, $4.0 \times 10^5$, $2.0 \times 10^5$, $1.0 \times 10^5$, $4.0 \times 10^4$, and $2.0 \times 10^2$, respectively (Fig. 8, Supplementary Fig. 6). A commercial fluorescence instrument of FLS920 was also employed to conduct the test, and a similar result was obtained (Supplementary Fig. 7).

Reusability is an unavoidable factor to be considered if an analytical technique is to be used practically. For this reason, both the reusability and the reversibility of the films, especially film 1, have to be examined. As depicted in Supplementary Fig. 4, the recovery of film 1 is the most difficult for magu. Accordingly, magu was chosen as an example of illicit drugs. To be reliable, more than 100 repetitive tests were conducted, and

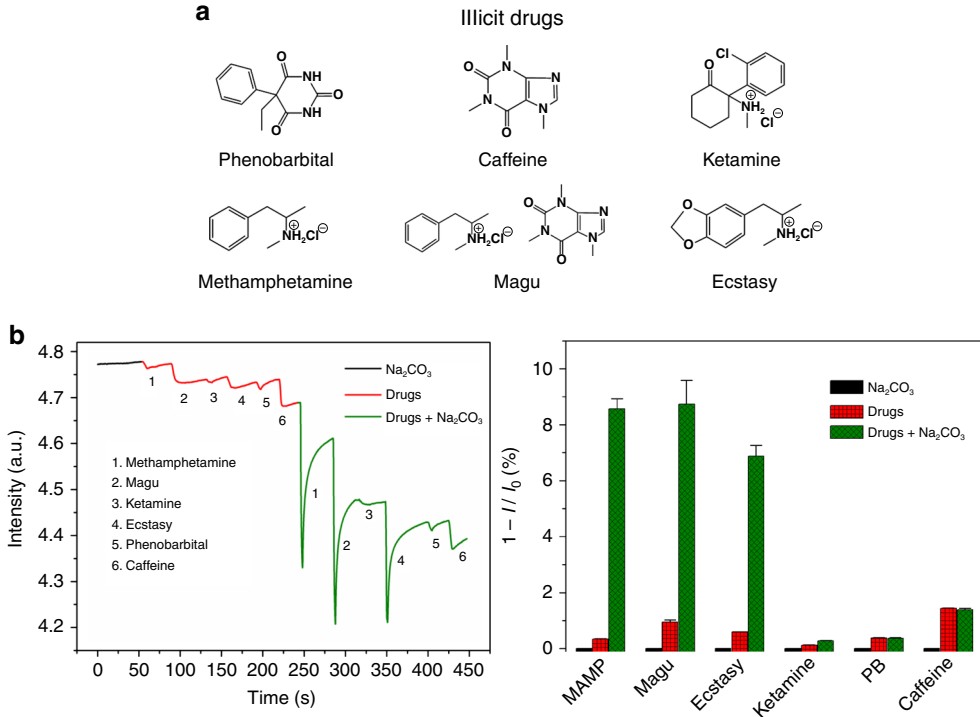

**Fig. 7** Illicit drug detection under different condition. **a** The chemical structures of the drugs. **b** Fluorescence responses of film 1 to the drug vapors in the presence or absence of $Na_2CO_3$. **c** Comparison of relevant responses. Clearly, introduction of the base enhances the response of the film to the drugs, especially to MAPA, magu, and ecstasy, of which the measurements were conducted for five times and the error bars are drawn from the maximum responses

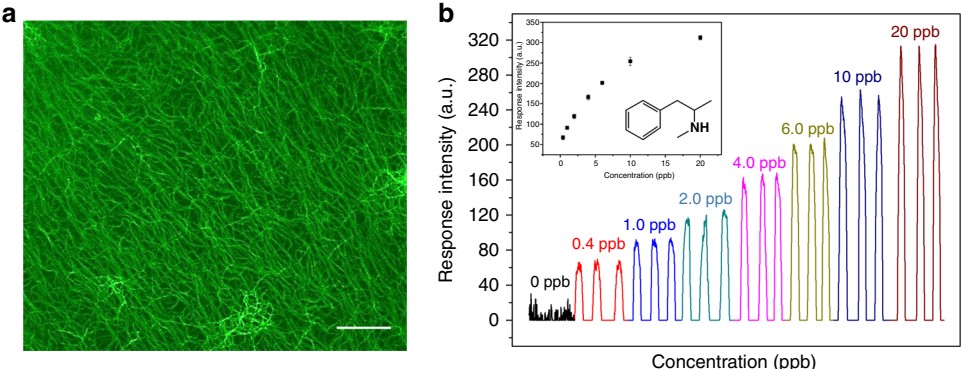

**Fig. 8** Micro-structure of film 1 and its responses to MAPA. **a** Fluorescence microscope image of the film ($\lambda_{ex} = 488$ nm). **b** Computer displayed results from the measurements at different MAPA vapor concentrations monitored with the homemade device (the saturated vapor pressure of MAPA is ~220 ppm at 25 °C[16]). Note: (1) each measurement was repeated three times; (2) inset is a plot of the average response to the vapor concentration, where the error bars are drawn from maximum to minimum responses

the results are presented in Supplementary Fig. 8. To examine the reusability, 10 successive tests, of which each lasted for 1 min, were performed first. It is found that the slow recovery of the fluorescence emission of the film shows no significant effect upon the detection provided the baseline is corrected from time to time (the inset of the figure), which is not hard to realize via signal processing. To examine if the emission is recoverable, the sample chamber was purged with air for more than 10 min at room temperature after each successive detection. It is clearly seen that the emission of the film was fully recovered, confirming the recoverability and reusability of the film.

In conclusion, we have developed a geometrical PBI dimer (PBI-CB), and it shows strong fluorescence in solid state. Further studies demonstrated PBI-CB-based films with either glass plate or plastic plate as substrate showed unprecedented sensing capability to the vapors of multiple illicit drugs, even if they exist in salt forms, which is beneficial for practical applications. Moreover, the potential interferences examined showed limited effect upon the sensing, and the effect can be ruled out by simply using the information provided by film 2 or film 3. Combination of the static and dynamic information obtained from the sensing tests could readily discriminate the illicit drugs under examination. Based upon the discoveries, a prototype two-sensor-based fluorescent illicit drug detector was developed. We believe that the two-step judgment method and the relevant detector as developed may find important applications in fast search of hidden illicit drugs.

## Methods

**Materials and instrumentations**. Tricosan-12-one (>98.0%), decaborane (United Boron, >95.0%), and 3, 4, 9, 10-perylenetetracarboxylic dianhydride (J&K Chemicals, >98.0%) were obtained commercially. Unless stated otherwise, all other reagents were obtained from commercial sources and used without further purification. Toluene and tetrahydrofuran were distilled over sodium in the presence of benzophenone under nitrogen atmosphere before use. Water used throughout was obtained from a Milli-Q reference system. The illicit drugs were provided by the Department of Public Security of Shaanxi Province, and used directly without further purification. (Note: the illicit drug samples used are all seized drugs by the local police officers from the illicit market.) $^{1}$H Nuclear magnetic resonance (NMR), $^{11}$B NMR, and $^{13}$C NMR spectra were obtained on a Bruker AV 600 NMR spectrometer. Pressed KBr disks for the powder samples were used for Fourier-transform infrared spectroscopy (FTIR) measurements, and the relevant spectra were obtained with a Bio-Rad FTIR spectrometer. The mass spectrometry data were collected on a Bruker maxis MALDI-TOF mass spectrometer in electron spray ionization-positive mode. Steady-state fluorescence excitation and emission spectra were obtained by using a time-correlated single photon counting fluorescence spectrometer (Edinburgh Instruments FLS920) with xenon lamp as the light source at room temperature.

**Fabrication of the sensing films**. According to literatures published recently[61], with inspection of the structure, it is anticipated that PBI-CB may form structured aggregates in suitable solvent or solvent mixtures through self-assembling driven by $\pi$–$\pi$ stacking and hydrophobic interaction. Thus, dioxane and water were separately chosen as a good solvent and a poor solvent to regulate the self-assembly behavior of the compound to achieve ordered structures. Specifically, to the dioxane solution of PBI-CB ($5.0 \times 10^{-5}$ mol L$^{-1}$), water was added as a poor solvent. Three hours later, small aggregates were formed in a dust-free container, which was used for film fabrication. A quantitative amount (20 μL) of the solution as prepared was added onto a substrate surface, which could be plastic plate (polypropylene, 8.0 mm/0.2 mm, radius/ thickness), bare glass (silicon dioxide, 8.0 mm/1.0 mm, radius/thickness), or silica-gel plate (silica gel, 400 meshes, 8.0 mm/0.2 mm, radius/thickness), at ambient temperature. In this way, three fluorescent films were obtained after evaporation of the solvents in air.

**Sensing measurements**. The sensing tests were carried out on the homemade device, where the sampling can be made automatically and the fluorescence intensity of the films, which were made into devices first as depicted in Fig. 4b, is monitored continuously, and the three films had been interrogated one after another. First hand vapors of the illicit drugs and potential interferences were used as analytes. For tests of the inorganic base-treated drugs, an equal amount of Na$_2$CO$_3$ and the drugs under tests were added to a bottle, then grinded with a glass rod for 10 min and finally, the vapor above the sample was collected for tests.

**Data availability**. All data supporting this study and its findings are available within the article and its Supplementary Information or from the corresponding author upon reasonable request.

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

## Acknowledgements

This work was supported by the Natural Science Foundation of China (21527802, 21673133, and 21603138), 111 project (B14041), Program for Changjiang Scholars and Innovative Research Team in University (IRT-14R33), and Science & Technology Research Program of Shaanxi Province (2017JQ2024). Ms Xiaomin Chen and Ruijuan Wen are thanked for helps in the development of the sensing platform and in performing some sensing tests. The authors would like to thank the Department of Public Security of Shaanxi Province for providing the illicit drug samples.

## Author contributions

Y.F. conceived the experiments, supervised the research work, and wrote the paper. K.L. synthesized the dimer PBI-CB, and carried out sensing experiments and helped analyzing the data and writing manuscript. C.D.S., Z.L.W., and Y.Y.Q. performed quantum chemical DFT calculations, and helped to conduct some experimental works. R.M., K.Q.L., and T.H.L. helped to analyze the sensing data, writing the manuscript, and performed the TEM measurements. All authors contributed to the interpretation of the results, and have given approval to the final version of the manuscript.

## Additional information

**Competing interests:** The authors declare no competing interests.

