## [Peer Review File · Nature Communications]

Reviewers' comments:

Reviewer #1 (Remarks to the Author):

I have read with keen interest the Nature Communications manuscript NCOMMS-17-29156A-Z entitled "Non-contact Identification and Differentiation of Illicit Drugs Using Fluorescent Films" submitted by Dr Fang et al.

This manuscript describes results of experiments on sensing vapors of illicit drugs using an optical sensing device comprising fluorescent films of bis(perylene bisimide)-o-carboranes. The films (and the resulting device) display fluorescence quenching in the presence of drugs, a signal, which can be further statistically evaluated to identify the analytes.

This manuscript describes a nice study and interesting results that could be potentially publishable in NatComm. Prior to publication, however, the Authors should devote a significant attention to re-writing the manuscript and dramatically improve the language. Further, I have some concerns and observations.

The manuscript is highly empirical and somewhat vague in details (...forming nice nano-fibers in suitable organic medium...). Reasonable amount of details should be provided.

Insufficient characterization of the deposited materials/films (TEM/diffraction measurements are not enough). Surface characterization (porosity) should be performed.

The three films – do they need to be interrogated one after another (serial) or are they evaluated simultaneously to yield pattern, which is then used to identify the drug? The manuscript seems to suggest that the analysis and the device always uses only one sensing film at the time. If true, this would invalidate the claim of fast response and analysis.

The Authors state that the sensor films display a differential behavior to the presence of the drug and interferent vapors. However, Film 1 and 2 showed drug-mediated quenching, while Film 3 no significant effect upon the film's emission, but the interferences showed quenching. Such similarity of the response may not bode well for the reliable classification of similar analytes. This may, very well stem from the electron transfer between the amine (drug) and the perylene diimide excited state as proposed. This would explain the similarity in the response for the drugs. At the same time, because of the general property of the amine as an electron donor, this method would be also sensitive to other amine compounds (benzylamine), legal drugs (e.g. Ritalin, dextromethorphan), tryptamines such as serotonin, food supplements such as aminoacids, etc. Somewhat bothersome is also the dynamics of the response, which, in some instances "no significant fluorescence recovery was observed". This would, in my opinion, lead to permanently biased sensor.

In summary, this manuscript describes an interesting topic and provides intriguing results. However, significant changes are required prior to publication.

Reviewer #2 (Remarks to the Author):

In this work, three film-based fluorescent sensors which sense the existence of six widely abused illicit drugs, were developed. The concept is interesting but it cannot be accepted before clarification of some issues. First, the interaction mechanism should be clearly identified. What is the basis of selectivity towards these drugs? What about the other compounds having amido bonds? On the other hand, water, shampoo, cream, hair conditioner, apple, pear, banana, and dirt clothes were classified as interferences, but actually these are various matrices not interferences compounds. What about the selectivity, it should be compared and evaluated also in the presence of other illicit selected drugs. In my opinion, a major revision is required.

Reviewer #3 (Remarks to the Author):

The research is outstanding. However, additional information/research would be welcome. What is the advantage of using the proposed fluorescence films based method over other previously proposed sensors (for instance similar research by the authors, reference 9)?.

- 1) Introduction is long and published reviews on gas vapour sensing would be welcome.
- 2) Although a picture regarding the home-made conceptual detector is shown in Figure 4, information regarding how the several films are placed/removed in the device is needed. In addition, are the three fluorescence films placed in the device at the same time, or is the sensing performed sequentially?. In addition, have the optimization/testing experiments be performed by exposing the films to saturated illicit drug/interference vapours? Is the device's response readable under non-saturated vapours conditions?
- 3) Drugs (vapours) are sampled by an air pump (Figure 4). What is the sampling flow rate?. At this point, how can the limit of detection be assessed?.
- 4) Specificity/selectivity studies. There are several illicit drugs which are chemical-structurally similar to MAPA (similar vapour pressure). The proposed sensing system should be tested for these drugs.

5) Re-usability. Can the films be re-used?

6) Applicability. Has the device been used in situ (on the field, under non-saturated vapours conditions)?

Responses to Reviewers:

To Reviewer 1

Comments:

I have read with keen interest the Nature Communications manuscript NCOMMS-17-29156A-Z entitled "Non-contact Identification and Differentiation of Illicit Drugs Using Fluorescent Films" submitted by Dr Fang et al.

This manuscript describes results of experiments on sensing vapors of illicit drugs using an optical sensing device comprising fluorescent films of bis(perylene bisimide)-o-carboranes. The films (and the resulting device) display fluorescence quenching in the presence of drugs, a signal, which can be further statistically evaluated to identify the analytes.

This manuscript describes a nice study and interesting results that could be potentially publishable in NatComm. Prior to publication, however, the Authors should devote a significant attention to re-writing the manuscript and dramatically improve the language. Further, I have some concerns and observations.

Response:

We thank the reviewer for the positive comments of our research. As required, the manuscript was re-written, and the language was polished by a native English speaker. We hope that our modified manuscript could satisfy the reviewer.

Q-1

The manuscript is highly empirical and somewhat vague in details (...forming nice nano-fibers in suitable organic medium...). Reasonable amount of details should be provided.

R-1

Thanks for your kind reminding. Description of the film structures was improved, and the details can be found at the first paragraph, page 4 of the revised manuscript. In addition, improvement of other descriptions or discussions was also made. As examples, see: paragraph 3 of page 5, paragraph 1 and 2 of page 6, and the last paragraph of page 12 of the revised manuscript.

Q-2

Insufficient characterization of the deposited materials/films (TEM/diffraction measurements are not enough). Surface characterization (porosity) should be performed.

R-2

As described in the manuscript, the films were fabricated by simple transfer of the aggregates of the sensing fluorophore, PBI-CB, onto different substrates surfaces. Accordingly, the morphologies of the films as fabricated are mainly determined by the structures of the aggregates produced in solution. Because of the reasons, systematic interrogation of the aggregate structures of PBI-CB at different stages in the mixed solvents was performed, and the results are depicted in Figure 2, and Supplementary Figure 2. It is seen that the final fibrous structures are formed via a route of nano-particles first, then fibers via amalgamation of the particles, and finally, the fibers aggregated into the structures as depicted.

As for the structures of the films, fluorescent microscopy studies were performed, and the results are shown in Figure 9 of the modified manuscript and Supplementary Figure 3 of the modified Supplementary Information. Clearly, both Film 1 and Film 2 possess fibrous structures no matter plastic or glass plate is used as substrate. For Film 3, however, its morphology is very different from the one observed on Film 1 and Film 2. The reason behind might be attributable to the roughness of the substrate (silica-gel plate) surface. This is because the capillaries on the substrate surface must slow down evaporation of solvents in the PBI-CB suspension which provides the molecules of the fluorophore a chance to re-organize in a new environment, and thereby forms new structures with solvent evaporation. Of course, the structure of the substrate surface also alters the morphology of the film observed.

To confirm the lamellar structure as revealed by TEM, SEM, and electron diffraction studies, X-Ray diffraction (XRD) studies were conducted. The result is shown in Supplementary Figure 2b of the modified Supplementary Information. The first two sharp signals follow the expected 1:1/2 ratio of the relevant d values, confirming the tentative result from TEM studies. The additional studies, relevant results and discussions are added to the revised manuscript (paragraph 1 and 2 of page 6) and the revised Supplementary Information (Supplementary Figure 2 and 3).

As for the porosity of the films, we are sorry for the unstrict statement. Actually, it is just a speculation as PBI-CB possesses a non-planar structural moiety of *o*-carborane which may screens dense packing at a molecular level. However, we have not got direct evidence for confirming the statement even though we have tried very hard. However, this does not mean the films as fabricated will have no chance to show the sensing properties as expected. This is

because the fibrous structures of the films, in particular Film 1 and Film 2, can also endow them with great surface area, which is a factor enhancing sensing sensitivity and sensing speed.

Q-3

The three films-do they need to be interrogated one after another (serial) or are they evaluated simultaneously to yield pattern, which is then used to identify the drug? The manuscript seems to suggest that the analysis and the device always uses only one sensing film at the time. If true, this would invalidate the claim of fast response and analysis.

R-3

Thank you very much for your valuable comments that helps us to improve the work. As indicated in your comments, there should be no doubt that collective using of the films at the same time is a preferential way as sensing speed could be increased and sensing operation could be simplified. But unfortunately, our tests reported in the original manuscript were conducted on a homemade, single-sensor based sensing platform. As far as we know, modification of the system into a three sensors-based new sensing platform is a big project, which is almost a technically impossible task to complete within a few months. This is because not only new hard-wares and new soft-wares need to be designed and produced, but also the structures, the soft-wares, and the operation parameters, especially the sampling unit, the optical unit and the signal processing unit of the sensor array, need to be optimized.

However, with careful examination of Figure 5 and Figure 6, it is clearly seen that the information obtained from Film 2 is just used for confirmation of the judgement acquired in the first step of the two-step strategy, and analysis in the second step utilizes only the information obtained from Film 1. In other words, fast response and analysis could be realized with collective using of Film 1 and Film 3 just, which means a two-sensor based sensing platform would play the function as expected. To our lucky, we do have a conceptual two-sensor based sensing platform (Supplementary Figure 5, the modified Supplementary Information), which was developed in our laboratory recently for other purpose. With utilization of this platform, some of the tests were re-conducted. The results from tests of three typical illicit drugs and three commonly found interferences are presented in the same figure. Clearly, the results are in consistence with those obtained independently by sequential using of the two films (Figure 3b, 3d of the revised manuscript). In this way, a fast discrimination and analysis was realized. As a

support to the conclusion obtained by sequential interrogation of the films as developed, the results and discussion from these additional tests are added to the revised manuscript (c.f. paragraph 2 of page 10, paragraph 1 and 2 of page 11) and the revised Supplementary Information (Supplementary Figure 5).

Q-4

The Authors state that the sensor films display a differential behavior to the presence of the drug and interferent vapors. However, Film 1 and 2 showed drug-mediated quenching, while Film 3 no significant effect upon the film's emission, but the interferences showed quenching. Such similarity of the response may not bode well for the reliable classification of similar analytes. This may, very well stem from the electron transfer between the amine (drug) and the perylene diimide excited state as proposed. This would explain the similarity in the response for the drugs. At the same time, because of the general property of the amine as an electron donor, this method would be also sensitive to other amine compounds (benzylamine), legal drugs (e.g. Ritalin, dextromethorphan), tryptamines such as serotonin, food supplements such as aminoacids, etc.

R-4

We understand that there are two concerns in this valuable comment. One is the classification ability of our method, and the second is the possible responses of the films to other compounds with a similar functionality that is amido-bond containing compounds.

As for the first concern, the classification was based upon a proposed two-step strategy. In the first step, positive or negative responses of the three films to a sample under examination were collectively employed to judge if it is an illicit drug or not. In the second step, information embodied in the response and recovery dynamics of Film 1 was semi-quantitatively extracted, and used to conduct further discrimination. As mentioned in the answer for question 3, information from Film 2 is employed just for confirmation.

As for the second concern, we agree with your comments. Emission from PBI derivatives are generally sensitive to presence of electron-rich compounds, especially organic amines, provided they are energetically matched. In other words, the organic compounds as mentioned in the comments are highly possible to induce positive response of the films. Of course, for films, response or not is not just determined by energetics. Substrate effect is another

unavoidable factor. Coming back to illicit drug detection, it is reasonable to consider real usage scenarios. For sniffing dogs, false alert is most commonly induced by vapors from toiletries, fruits and even dirty clothes, which are common substances in travel. But unfortunately, they are rarely mentioned in illicit drug detection studies.

To elaborate your concern, some amido-bond containing organic compounds, including three organic amines (benzylamine, ethylhexylamine, tryptamine), four legal drugs (atorvastatin calcium tablets, dextromethorphan, piperidine drugs (substituted as piperidine), and dopamine), and five typical amino-acids of different kinds (glycine, tryptophan, lysine, glutamic acid, and histidine), were chosen as additional interferences. Sensing tests demonstrated that of the 12 tested compounds, only benzylamine, ethylhexylamine and piperidine showed significant effect upon the fluorescence emission of the three films (Figure 3 and Figure 5, the modified manuscript). As depicted in the figures, the response patterns of these compounds are also different from those of the illicit drugs under tests. Differentiation with the two-step strategy can be still fulfilled. More importantly, the information obtained from Film 2, again, is still just used for confirming the judgement made by using Film 1 and Film 3. The results as obtained were added to the modified manuscript, and the details can be found at paragraph 1 and 3 of page 7, paragraph 1, 2 and 4 of page 8 of the revised manuscript, and Supplementary Figure 5 of the revised Supplementary Information.

Q-5

Somewhat bothersome is also the dynamics of the response, which, in some instances “no significant fluorescence recovery was observed”. This would, in my opinion, lead to permanently biased sensor.

R-5

The statement as mentioned is a description of the observation as obtained within the first minute after removing the sample under examination. This observation was used for fast discrimination, but not a real reflection of the recovery process. As described in the main text, the recovery of Film 1 is most difficult for magu. To exam the reusability and reversibility of the film, magu was chosen as an example of illicit drugs. With the analyte, more than 100 tests were conducted, and the results are presented in Supplementary Figure 8 of the revised Supplementary Information. To exam the reusability, 10 successive tests, of which each lasted

for 1 minute, were performed first. It is found that the slow recovery of the fluorescence emission of the film shows no significant effect upon the detection provided the base-line is varied from time to time (the inset of the figure), which is not hard to realize via signal processing. To exam if the emission is recoverable, the sample chamber was purged with air for more than 10 minutes at room temperature after each successive detection. It is clearly seen that the emission of the film was fully recovered, confirming the recoverability and reusability of the film. This information was added in the revised manuscript (c.f. paragraph 2 of page 14) and the revised Supplementary Information (c.f. Supplementary Figure 8).

Q-6

In summary, this manuscript describes an interesting topic and provides intriguing results. However, significant changes are required prior to publication.

R-6

Thank you for your positive comments and interest to our work. As required, all the changes have been made with care.

We hope our efforts have addressed all your concerns well.

To Reviewer 2

Comments:

In this work, three film-based fluorescent sensors which sense the existence of six widely abused illicit drugs, were developed. The concept is interesting but it cannot be accepted before clarification of some issues. First, the interaction mechanism should be clearly identified. What is the basis of selectivity towards these drugs? What about the other compounds having amido bonds? On the other hand, water, shampoo, cream, hair conditioner, apple, pear, banana, and dirt clothes were classified as interferences, but actually these are various matrices not interferences compounds. What about the selectivity, it should be compared and evaluated also in the presence of other illicit selected drugs. In my opinion, a major revision is required.

Response:

We thank the reviewer for the positive comment to our work. Here are our answers to the concerns raised by the reviewer:

Theoretical calculations revealed that the quenching of the illicit drugs as examined to the fluorescence emission of the films can be attributed to the electron transfer from the drugs, which are electron-rich ones, to the HOMO orbital of the sensing fluorophore, a derivative of PBI, typical electron-poor structure, when it is excited, a typical photo-induced electron transfer (PET) phenomenon (c.f. page 11 of the revised manuscript, and Figure 7).

The selectivity of the films to the interferences may originate from the un-matched energetics and the screening effect of the substrates. Additional descriptions and discussions can be found at paragraph 3 of page 12, and paragraph 3 of page 13 of the revised manuscript.

Water was employed as an interference as it is a well-known factor to affect the emission of fluorescence films. The vapors from shampoo, cream, hair conditioner, apple, pear, banana, and dirty clothes induce false alert frequently, when sniffing dogs were employed to search hidden drugs and/or explosives. Unfortunately, this is rarely mentioned in the literatures, but the police officers know the situation. The final purpose of our study is to develop a technique which is expected to be practically usable. For this reason, real usage scenarios have to be taken into consideration, which explains why water, shampoo, cream, hair conditioner, apple, pear, banana, and dirty clothes were chosen as interferences.

As for the effect of other amido-bond containing organic compounds, three organic amines (benzylamine, ethylhexylamine, tryptamine), four legal drugs (atorvastatin calcium tablets, dextromethorphan, piperidine drugs (substituted as piperidine), and dopamine), and five typical amino-acids of different kinds (glycine, tryptophan, lysine, glutamic acid, and histidine), were chosen as additional interferences. Sensing tests revealed that of the 12 tested compounds, only benzylamine, ethylhexylamine and piperidine showed significant effect upon the fluorescence emission of the three films (Figure 3 and Figure 5, the modified manuscript). As depicted in the figures, the response patterns of these compounds are also different from those of the illicit drugs under tests. Differentiation with the two-step strategy can be still fulfilled. More importantly, the information obtained from Film 2, again, is still just used for confirming the judgement made by using Film 1 and Film 3. The results as obtained were added to the modified manuscript, and the details can be found at paragraph 1 and 3 of page 7, paragraph 1, 2, and 4 of page 8, paragraph 2 of page 10, and paragraph 1 and 2 of page 11 of the revised manuscript, and Supplementary Figure 5 of the revised Supplementary Information.

We are sorry that our method can't be used for selective detection of an illicit drug in the

presence of other detectable drugs, a phenomenon frequently encountered in chemical sensor reported in literatures (*J. Am. Chem. Soc.* **2012**, 134, 4834; *Angew. Chem. Int. Ed.* **2017**, 56, 9860; *Adv. Mater.* **2017**, 29, 1604528). The pre-requirement for discrimination reported here and those in literatures is that the sample is not a mixture of the detectable analytes. But selective detection as expected would be realized if sensor arrays with more powerful discrimination abilities were developed. The effort in this regard is in progress.

We hope our efforts have addressed all your concerns well.

To Reviewer 3

Comments:

The research is outstanding. However, additional information/research would be welcome. What is the advantage of using the proposed fluorescence films based method over other previously proposed sensors (for instance similar research by the authors, reference 9)?.

Response:

Thanks for your positive comment, encouragement and precious suggestions. As for the detection of hidden illicit drugs via vapor sensing, the performance of our method reported in this manuscript is much superior than those reported earlier as evidenced by the following facts: first, only MAPA can be detected by the method reported earlier, not to mention that most of the tests were conducted with a simulant of the drug that is *N*-methyl-phenethylamine (MPEA); second, for the present method pre-concentration and pre-treatment of the drug samples are not necessary owing to its unprecedented sensitivity; third, for the present study, the tests were conducted on a homemade sensing platform, which lays foundation for practical applications. Therefore, we are confidence to conclude that our method reported in the present work stands for a milestone in the detection of hidden illicit drugs. Corresponding descriptions and discussions can be found at paragraph 1 of page 3, paragraph 2 of page 7, paragraph 2 of page 10, and paragraph 1 and 2 of page 11, respectively, of the revised manuscript.

Q-1

Introduction is long and published reviews on gas vapour sensing would be welcome.

R-1

The introduction has been simplified and polished as required. Suggestion on preparation of a review will be seriously considered.

Q-2

Although a picture regarding the homemade conceptual detector is shown in Figure 4, information regarding how the several films are placed/removed in the device is needed. In addition, are the three fluorescence films placed in the device at the same time, or is the sensing performed sequentially? In addition, have the optimization/testing experiments be performed by exposing the films to saturated illicit drug/interference vapours? Is the device's response readable under non-saturated vapours conditions?

R-2

Our responses to the concerns raised by the reviewer are as follows: (1) The information as required has been added, and the details can be seen at paragraph 2 at page 16 of the revised manuscript; (2) The results reported in the original manuscript were obtained by sequential using of the films as developed, but to be more practical, a two-sensor based sensing platform was developed and employed for re-conduction of the tests of three typical illicit drugs and three commonly found interferences. This sensing platform and the results as obtained are presented in Supplementary Figure 5. Clearly, the results are in consistence with those obtained independently by sequential using of the two films (Figure 3b, 3d of the revised manuscript). In this way, a fast discrimination and analysis was realized. As a support to the conclusion obtained by sequential interrogation of the films as developed, the results and discussion from these additional tests are added to the revised manuscript (c.f. at paragraph 1 and 3 of page 7, paragraph 1, 2 and 4 of page 8 of the revised manuscript,) and Supplementary Figure 5 of the revised Supplementary Information; (3) Yes, the tests can be performed at much lower concentrations of the drug vapors. As reported in the manuscript, detection limits for the six relevant illicit drugs must be significantly lower than 5.0×10^5 , 4.0×10^5 , 2.0×10^5 , 1.0×10^5 , 4.0×10^4 and 2.0×10^2 times respective dilution of the saturated vapors of them at room temperature (c.f. paragraph 4 of page 13, and paragraph 1 of page 14 of the revised manuscript).

Q-3

Drugs (vapours) are sampled by an air pump (Figure 4). What is the sampling flow rate? At this

point, how can the limit of detection be assessed?

R-3

The sampling flow rate is ~3.0 mL/s. Detection limits of the method to the drugs under examination were determined by using a static volumetric method. Specifically, suctioning a certain volume of the saturated vapor of the drug with an empty glass-injector, then suctioning fresh air into the glass-injector so that the gas volume increased to a certain value. In this way, the drug vapor was diluted. Further dilution can be realized by squeezing out some of the mixture vapors, and then suctioning fresh air again. The diluted vapors as generated were used for detection limit determination. Relevant revisions can be found at paragraph 4 of page 13, and paragraph 1 of page 14 of the revised manuscript.

Q-4

Specificity/selectivity studies. There are several illicit drugs which are chemical-structurally similar to MAPA (similar vapour pressure). The proposed sensing system should be tested for these drugs.

R-4

To be honest, we have not got a license to work with illicit drugs in our lab yet. The tests reported in this work were conducted with the aid of the Department of Public Security of Shaanxi Province. All illicit drugs available locally were taken as sample analytes in our studies. Anyway, we will try to get more illicit drug samples for tests in the future.

Q-5

Re-usability. Can the films be re-used?

R-5

This concern is of great values for practical applications. To clarify the concern, additional tests were conducted. As described in the main text, the recovery of Film 1 is most difficult for magu. To exam the reusability and reversibility of the film in the detection, magu was chosen as an example of illicit drug. To be reliable, more than 100 repetitive tests were conducted, and the results are presented in Supplementary Figure 8. To exam the reusability, 10 successive tests, of which each lasted for 1 minute, were performed first. It is found that the slow recovery of the fluorescence emission of the film shows no significant effect upon the detection provided the base-line is varied from time to time (the inset of the figure), which is not hard to

realize via signal processing. To exam if the emission is recoverable, the sample chamber was purged with air for more than 10 minutes at room temperature after each successive detection. It is seen clearly that the emission of the film was fully recovered, confirming the recoverability and reusability of the film. This information was added to the revised manuscript (c.f. paragraph 2 of page 14) and the revised Supplementary Information (c.f. Supplementary Figure 8).

Q-6

Applicability. Has the device been used in situ (on the field, under non-saturated vapours conditions)?

R-6

Not yet. Field tests will be performed after a prototype detector is built. To upgrade our conceptual device into a practically usable prototype detector, more technical and engineering work needs to be conducted. However, the future is very bright as our device can be used for the detection of very dilute vapors of the illicit drugs, of which positive detection can be still performed after 5.0×10^5 , 4.0×10^5 , 2.0×10^5 , 1.0×10^5 , 4.0×10^4 and 2.0×10^2 times respective dilution of the saturated vapors of the six illicit drugs at room temperature.

We hope our efforts have addressed all your concerns well.

Reviewers' Comments:

Reviewer #2 (Remarks to the Author):

I am convinced about the revisions and required corrections. It can be accepted as it is.